Drought and water availability analysis for irrigation and household water needs in the Krueng Jrue sub-watershed

Basri Hairul hairulbasri@unsyiah.ac.id 1
Sufardi Sufardi 1
Helmi Helmi 1
Syakur Syakur 1
Sugianto Sugianto 1
Azmeri Azmeri 2
Helmi Helmi 3
1 Soil Science Department, Faculty of Agriculture, Universitas Syiah Kuala , Banda Aceh , Aceh , Indonesia
2 Civil Engineering Department, Faculty of Engineering, Universitas Syiah Kuala , Banda Aceh , Aceh , Indonesia
3 Forestry Department, Teungku Chik Pante Kulu, School of Forestry Science , Banda Aceh , Aceh , Indonesia
Meraj Gowhar
Electronic publication date: 2023 Feb 9
Publication date: 2023
Volume: 11
Electronic Location ID: e14830
Received 2022 Sep 16; Accepted 2023 Jan 9
Copyright: ©2023 Basri et al.
Copyright year: 2023
Copyright holder: Basri et al.
License: This is an open access article distributed under the terms of the Creative Commons Attribution License, which permits unrestricted use, distribution, reproduction and adaptation in any medium and for any purpose provided that it is properly attributed. For attribution, the original author(s), title, publication source (PeerJ) and either DOI or URL of the article must be cited.
License URL: https://creativecommons.org/licenses/by/4.0/

Keywords: Z-score statistics, Mock model, Water availability, Water needs, Agricultural water management, Watershed management, Drought analysis

Funding: The authors received no funding for this work.

==============================
This study aimed to analyze drought conditions and evaluate irrigation water availability and household water needs in the Krueng Jrue sub-watershed, Aceh Province, Indonesia. The Z-score statistics method was developed to analyze the drought, and the Mock model was used to generate discharges. We performed model validation using linear regression, which produced a coefficient of determination (R2 = 0.90**) and coefficient of regression (r = 0.95**). In general, this area had a normal Z-score for precipitation (ZSP) class with 90 events (75%) and a normal Z-score for a discharge (ZSD) class with 89 events (74.2%). There were 0–11 (0–9.2%) moderate wet, very wet, extreme wet, moderate drought, and severe drought events. The consistency between the ZSP and ZSD indices reached 85.8%, indicating consensus between the meteorological droughts that were analyzed based on rainfall (ZSP) and hydrological droughts analyzed based on water discharge (ZSD). ZSP and ZSD indices showed negative values during the dry season (April to September) and positive values during the rainy season (October to March). There was a surplus of water availability for irrigation and household water needs during the rainy season and a deficit during the dry season. However, water deficits also occurred in certain months during the rainy rendeng planting season, for example, in October 2009, 2013, 2016, and 2017 as well as in February between 2008 to 2011 and from 2014 to 2017. This observation was probably due to the influence of global climate variables that need to be substantiated. This study offers necessary information for farmers, the community, and the local government when anticipating drought phenomenon, organizing the rice planting season, and evaluating water availability in other watersheds.

Introduction

Climate change impacts drought, floods, and water scarcity. Climate issues require an understanding of hydrological dynamics and trends based on current watershed biophysical conditions in order to ensure better preparedness and response. One of the most essential activities needed to anticipate water scarcity is the evaluation of meteorological and hydrological drought and water availability based on the current biophysical conditions of a watershed.

Although a hydrological disaster cannot be avoided, it can be anticipated using scientific and technological development supported by accurate data (Tallaksen, Hisdal & Van Lanen, 2009) in order to minimize environmental damage. This is determined using primary and additional components of the hydrological disaster vulnerability parameters (Lohani, Krishan & Chandniha, 2017), which stresses the importance of understanding an area’s description, the land’s biophysical characteristics, and the response to changes in the hydrological cycle due to global climate change and extreme weather (Van Huijgevoort et al., 2014). Preliminary studies conducted through analysis of 20 years of rainfall data using the Schmidt-Ferguson method in Aceh Besar District (including the Krueng Aceh sub-watershed) concluded that the climate type was B (wet) in 1980 but changed to C (slightly dry) in 2000 (Basri, Syahrul & Nursidah, 2002).

The Krueng Jrue sub-watershed is a part of the Krueng Aceh watershed located at its upper stream. The Krueng Aceh plays a vital role as the main water source for Aceh Besar District and Banda Aceh City, Indonesia. The increased intensity of land conversion negatively impacted the hydrological conditions of the Krueng Aceh watershed and caused an increase in peak discharge, discharge fluctuations between the dry and rainy seasons, runoff coefficients, as well as an increase in erosion, sedimentation, flooding, and drought (Nasrullah & Kartiwa, 2010).

Of Indonesia’s 108 priority handling watersheds, the Krueng Aceh is one of the most critical. The critical land area required in the Krueng Aceh watershed, mainly located in the Krueng Jrue sub-watershed, increased from 2,320.88 ha (10.00%) in 2013 to 10,969.85 ha (47.25%) in 2018. A decrease in the biophysical quality of a watershed can be caused by reduced land cover that can increase surface runoff or otherwise reduce soil infiltration capacity (Basri et al., 2022). Therefore, watershed sustainability can be achieved by identifying the links between land, hydrology, and the related upstream and downstream areas that affect the watershed and sub-watershed ecosystem units (Susetyaningsih, 2012). The water availability in the Krueng Jrue sub-watershed ranges from 0.24 to 3.22 m3s−1. The total water demand for households and irrigation is 0.18–6.44 m3s−1 (Isnin, Basri & Romano, 2012). However, this study did not specifically analyze drought indices, only the economic value of water in the Krueng Jrue sub-watershed.

Meteorological drought is usually defined by the degree of drought (compared to some “normal” or average number) and the duration of the dry period. Furthermore, hydrological drought is associated with the effect of precipitation periods on surface or subsurface water supplies. The frequency and severity of hydrological droughts are often defined on a watershed scale (Wilhite & Glantz, 1985).

Drought indices in a watershed can be analyzed using the Standard Precipitation Index (SPI) (McKee, Doesken & Kleist, 1993) and Standard Discharge Index (SDI) (David & Davidová, 2016). Discharges can be generated using a rainfall-runoff model called the Mock model, first introduced by FJ Mock to predict potential water availability (Mock, 1973). It is useful in predicting the occurrence of hydrological drought and utilizing water as efficiently as possible (Caraka et al., 2018). The water surplus or deficit can be evaluated based on potential water availability.

The SPI was first introduced using gamma distribution (McKee, Doesken & Kleist, 1993) and has been used by researchers in many different countries (Ceglar, Zalika & Lucka, 2008; Shah, Bharadiya & Manekar, 2015; Pathak & Dodamani, 2016; Jang, 2018; Liu et al., 2018; Naresh Kumar et al., 2009; Jiménez-Donaire, Tarquis & Giráldez, 2020). World meteorological organizations released an SPI guide, and the latest SPI program (SPI_SL_6.exe) is downloadable for free (Svoboda, Hayes & Wood, 2012). Some researchers also use the term SPI in Z-score statistics to examine the abnormal occurrence of rainfall or discharges (Wu et al., 2001; Bhuiyan, RP & Kogan, 2006; Tsakiris & Vangelis, 2004; Khan, HF & Rana, 2008; Omonijo & Okogbue, 2014; Dogan, Berktay & Singh, 2012; Jain et al., 2015; Suribabu & Sujatha, 2019; Li et al., 2019). The Z-score statistical method was developed to analyze drought due to the simplicity of this method. Furthermore, the Mock model was used to obtain discharge data based on water balance analysis. The Mock model is generally used in Indonesia, especially on the island of Java, and existing references suggest that the Mock model has never been used to analyze water availability for a watershed/sub-watershed in other countries. Therefore, this Mock model can be tested as an alternative rainfall-runoff model for watersheds/sub-watersheds in other countries by adjusting the parameters.

A study on drought analysis using the Z-score statistical method and Mock models to evaluate water availability in the Krueng Jrue Sub-Watershed has never been conducted. This study aims to analyze drought and evaluate the availability of water for irrigation needs and household water needs in the Krueng Jrue sub-watershed, Aceh Province, Indonesia. Furthermore, this study is expected to find new crucial information regarding: (1) drought conditions and the consistency between meteorological drought and hydrological drought based on rainfall data and discharge data of a watershed, (2) water availability conditions for irrigation and household water demand during the rainy rendeng growing season and dry gadu planting season, and (3) rice and secondary crop planting schedules based on drought analysis and water availability.

Materials & Methods

Times and site

The research was conducted in 23,218.06 ha of the Krueng Jrue sub-watershed located at 5°12′–5°28′N and 95°20′–95°32′E, which is part of the Krueng Aceh watershed, Aceh Province, Sumatra, Indonesia (Fig. 1). It was conducted from January to December 2019.

Figure 1 Location of the study area in Krueng Jrue sub-watershed.

Data collection

Materials needed included maps (administrative, topography, soil type, and land use), and monthly rainfall, evapotranspiration, irrigation area, and population data for 2008–2017. A land map was obtained from the Krueng Aceh Watershed Management Board and climatology data were from Blang Bintang Meteorological, Climatological, and Geophysical Agency. While this study used data from the Indrapuri rainfall station, as shown in Fig. 1, monthly river discharge data were collected from the Center of River Basin Sumatera-I. There are three climatological stations in the Krueng Aceh watershed, but only one is located in the Krueng Jrue sub-watershed. Furthermore, the discharge data were collected from Kr. Meulesong hydrometry station using an automatic water level recorder (AWLR) that was set up in the field.

Drought indices

The Z-score for precipitation (ZSP) was used in this study to evaluate the meteorological drought, as shown in Eq. (1). (1) ZSP=Pi−PavgS,

where, Pi = precipitation (mm); Pavg = average of precipitation; S = standard deviation of precipitation.

Using the same concept, the Z-score for discharges (ZSD) was calculated to evaluate hydrological drought using the formula shown in Eq. (2). (2) ZSD=Di−DavgSd,

where, Di = discharge (m3s−1); Davg = average of discharge; Sd = standard deviation of discharge. Drought criteria to justify the drought class for ZSP and ZSD are shown in Table 1.

Table 1 Drought class for ZSP and ZSD.

No	Drought criteria	Values of ZSP and ZSD	
1	Extreme wet (EW)	≥ 2.00	
2	Very wet (VW)	1.50 to 1.99	
3	Moderate wet (MW)	1.00 to 1.49	
4	Normal (N)	−0.99 to 0.99	
5	Moderate drought (MD)	−1.00 to −1.49	
6	Severe drought (SD)	−1.50 to −1.99	
7	Extreme drought (ED)	≤ −2.00	
Notes.

Source: (Ceglar, Zalika & Lucka, 2008).

The Mock model

The basic approach of the Mock model is to consider factors such as rainfall, evapotranspiration, and water balance at the soil surface, as well as groundwater content. Monthly rainfall data are needed to analyze the river’s water availability and are the primary input of the Mock method. The longer the recording period, the better the results will be. Many researchers (Setyawan, Lee & Prawitasari, 2016; Putro, 2016; Sebayang & Trianing, 2018; Chandrasasi, Limantara & Juni, 2020; Maulana, Suhartanto & Harisuseno, 2019; Krisnayanti et al., 2019; Dinar, Agus & Sarino, 2020) have used the Mock model to assess water discharges or availability of watersheds. Generally, these studies found the Mock model suitable for evaluating the water balance in specific watersheds. The equations used to calculate water balance parameters by the Mock model (Mock, 1973; Umum, 1986) are as follows:

Evapotranspiration: (3) E=ET0−ΔE

(4) ΔE=ET0m1/2018−n1,

where, ΔE = the difference between potential and actual evapotranspiration (mm month−1); ET0 = potential evapotranspiration (mm month−1); m1 = the proportion of soil surface that is not covered by vegetation (set as 20%); n1 = total of rainy days; E = actual evapotranspiration (mm month−1).

River discharges: (5) Qriver=QtotalxA/t

(6) Qtotal=Qbase+Qdirect+Qstorm,

where, Qriver = discharges of a river (m3s−1), Qtotal = total runoff (mm month−1), A = watershed area (Ha), t = time (second) Qbase=baseflow (mm month−1), Qdirect = direct runoff (mm month−1), and Qstorm = storm runoff (mm month−1).

Baseflow: (7) Qbase=inf−G.STORt+G.STORt−1

(8) inf=WSxIF

(9) WS=ISM+Re−E−SMS

(10) SMS=ISM+Re−E

(11) G.STORt=G.STORt−1xRc+0.51+Rcxinf,

where, inf = infiltration (mm month−1); G. STORt = groundwater storage at the beginning of the month (mm month−1); G. STOR(t−1) = groundwater storage at the end of the month (mm month−1); IF = infiltration factor (set as 0.4); WS = water surplus (mm month−1); ISM = initial soil moisture (set as 200 mm month−1); Re = monthly rainfall (mm month−1); SMS = soil moisture storage (mm month−1); Rc = flow reduction coefficient (set as 0.6).

Direct runoff: (12) Qdirect=Wsx1−IF,

where Ws = water surplus (mm)

Storm runoff: (13) Qstorm=RexPF,

where, PF = precipitation factor (%).

Water demand

Two kinds of water needs were considered to calculate the water demand. First, water demand for irrigation in the Krueng Jrue sub-watershed between 2008–2017 was projected based on the area of irrigated land according to irrigation water needs, which was calculated as follows: (14) DR=NFRe×8.64,

where, DR = diversion requirement (l s−1ha−1); NFR = net water requirement in paddy field (l s−1ha−1); e = irrigation efficiency; 1/8.64 = conversion value from (mm day−1) to (l s−1ha−1).

Second, the water demand for households in the Krueng Jrue sub-watershed during 2008–2017 was calculated using the assumed population growth (1.4% year−1) and standard water demand per capita (0.06 m3day−1).

Model validation

Validation was performed by comparing the observation discharges (Qo) with the Mock model discharges (Qc) using linear regression. The statistical parameters were determined using the coefficient of determination (R2) and value of the regression coefficient (r). They should be near 1 in order to ensure that the Mock model is valid and suitable to predict discharges for this region. The correlation coefficient (r) can be calculated using the following equation (Ward & Trimble, 2003): (15) r=n∑QoQc−∑Qo∑Qcn ∑Qo2+ ∑Qo2.n∑Qc2− ∑Qc2.

Results

Model validation

Model validation was conducted using regression analysis to observe the correlation between the observed discharges and calculated discharges using the Mock model. Statistically, the model showed a very significant regression coefficient (r = 0.95**) and coefficient of determination (R2 = 0.90**). Those values prove the validation of the Mock model to predict discharges for this region. The correlation between observed discharges and calculated discharges is shown in Fig. 2.

Monthly rainfall

The average monthly rainfall in the Krueng Jrue sub-watershed (2008–2017) is shown in Fig. 3. Indonesia has two seasons: the rainy season (October–March) and the dry season (April–September). The highest monthly average rainfall occurred in December, November, and January at 325.7, 324.4, and 269.9 mm, respectively. The lowest monthly average rainfall occurred in February (98.3 mm) and July (68.7 mm). February is included in the rainy season, but the rainfall observed in February (98.3 mm) was less than the average rainfall observed for all of the months (190.3 mm). However, April and May are included in the dry season, but had average rainfalls of 247.4 and 219.6, respectively, which was above the average rainfall for all of the months.

Monthly discharges

Table 2 shows the discharges of the Krueng Jrue sub-watershed generated using the Mock model. The higher the rainfall, the higher the water discharge generated by the Mock model. The average monthly discharges of the Krueng Jrue sub-watershed from 2008–2017 ranged from 3.16–31.18 m3s1. The highest monthly average discharges during the rainy season (October–March) occurred in November (31.18 m3s−1) and December (28.02 m3s−1), while the lowest monthly discharge in the dry season (April–September) occurred in July (3.16 m3s−1), and June (3.36 m3s−1). Further analysis provided information for the entire observation year. February, which is included in the rainy season, had a meager monthly discharge value (1.78–3.76 m3s−1), that was below the average monthly discharge, except in 2012 and 2013.

Figure 2 Correlation between observed discharges and calculated discharges.

Figure 3 Average monthly rainfall in the Krueng Jrue sub-watershed (2008–2017).

Table 2 Discharges of the Krueng Jrue sub-watershed.

Month	Discharges (m 3 s −1 )	
	2008	2009	2010	2011	2012	2013	2014	2015	2016	2017	Total	Average	
Jan.	4.39	16.99	3.00	29.90	17.44	40.59	13.58	24.23	40.18	45.75	236.05	23.61	
Feb.	2.75	2.17	2.31	3.06	26.55	19.81	1.78	2.18	3.76	3.31	67.68	6.77	
Mar.	15.80	5.14	4.29	15.99	7.97	3.44	2.10	2.35	2.26	6.16	65.50	6.55	
Apr.	41.79	14.46	4.97	22.71	32.09	28.46	4.01	27.14	2.41	6.60	184.64	18.46	
May	2.69	13.13	15.81	2.76	5.02	28.77	5.82	6.71	2.67	14.84	98.22	9.82	
Jun.	6.86	2.42	8.82	2.17	1.73	2.90	2.05	1.92	2.51	2.20	33.58	3.36	
Jul.	3.50	2.19	9.17	2.40	2.82	2.18	1.64	2.56	2.89	2.24	31.59	3.16	
Aug.	5.65	2.96	8.44	3.81	2.97	2.23	3.43	6.00	3.11	3.25	41.85	4.19	
Sep.	14.47	2.97	43.03	6.65	2.52	2.64	3.79	4.93	2.55	15.90	99.45	9.95	
Oct.	5.01	4.25	6.30	16.93	7.41	2.75	19.21	23.30	2.99	3.81	91.96	9.2	
Nov.	37.45	38.85	44.32	11.74	30.30	5.84	46.68	39.95	25.19	31.51	311.83	31.18	
Dec.	11.31	21.25	44.96	23.34	38.60	45.33	41.90	13.72	10.73	29.04	280.18	28.02	
Total	151.67	126.78	195.42	141.47	175.42	184.94	145.99	154.99	101.25	164.61	1,542.53	154.27	
Average	12.64	10.57	16.29	11.79	14.62	15.41	12.17	12.92	8.44	13.72	128.54	12.86	

Table 2 shows that the average monthly discharge was 12.86 m3 s−1. Average monthly discharges for 2010, 2012, 2013, 2015, and 2017 were above the average monthly discharges for the whole year. On the contrary, the average discharges for 2008, 2019, 2011, and 2016 were below the average discharges. Based on the average discharges for each month, August had an average discharge below the average monthly discharge throughout the year. In the rainy season (October–March), the average monthly discharge should have been above the average monthly discharge for the whole year, but the average discharges for October, February, and March were below the average monthly discharge for the whole year. Conversely, for the dry season (April–September), the average discharge should have been below the monthly average discharge for the entire year, but the results of the average discharge for April were above the monthly average discharge for the entire year.

Z-score values for precipitation and discharge

The Z-score for precipitation (ZSP) and discharge (ZSD) from 2008 to 2017 are shown in Tables 3 and 4. Positive ZSP or ZSD values indicate greater than median precipitation or discharges and negative values indicate lower than median precipitation or discharges (Tsakiris & Vangelis, 2004). This region had drought indices for ZSP: 90 normal (N) events (75%), six moderate wet (MW) events (5%), seven very wet (VW) events (5.8%), six extreme wet (EW) events (5%), and 11 moderate drought (MD) events (9.2%). Drought indices for ZSP with severe drought and extreme drought were not found in this region. Furthermore, this region had drought indices for ZSD: 89 normal (N) events (74.2%), 11 moderate wet (MW) events (9.2%), four very wet (VW) events (3.3%), six extreme wet (EW) events (5%), nine moderate drought (MD) events (7.5%), and one severe drought (SD) event (0.8%). Drought indices for ZSD with extreme drought were not found in this region. The ZSP and ZSD indices in the dry season (April–September) tended to be negative, indicating that rainfall or discharge was under the average index. However, they tended to be positive for the rainy season (October–March).

Table 3 Z-score for precipitation (ZSP) values.

Month	ZSP	ZSP	ZSP	ZSP	ZSP	ZSP	ZSP	ZSP	ZSP	ZSP	
	2008	Criteria	2009	Criteria	2010	Criteria	2011	Criteria	2012	Criteria	2013	Criteria	2014	Criteria	2015	Criteria	2016	Criteria	2017	Criteria	
Jan.	−0.88	N	−0.28	N	−1.00	MD	−0.20	N	−0.02	N	0.67	N	−1.43	MD	0.12	N	1.46	MW	1.57	VW	
Feb.	−0.37	N	0.91	N	−0.72	N	−0.004	N	−0.04	N	0.95	N	−0.35	N	−0.40	N	0.56	N	2.65	EW	
Mar.	0.33	N	−0.09	N	−0.09	N	0.40	N	0.01	N	0.18	N	−0.38	N	−0.15	N	−0.41	N	3.02	EW	
Apr.	−0.78	N	−1.04	MD	−0.17	N	−0.28	N	−0.65	N	−0.54	N	−0.54	N	0.82	N	1.09	MW	2.09	EW	
May	−1.03	MD	−0.91	N	−0.12	N	−0.98	N	−0.52	N	0.27	N	−0.36	N	0.52	N	2.07	EW	1.06	MW	
Jun.	−0.32	N	−0.88	N	0.18	N	−0.97	N	−0.94	N	0.19	N	−0.38	N	0.82	N	2.33	EW	−0.03	N	
Jul.	−0.63	N	−1.16	MD	0.71	N	0.19	N	−0.18	N	−0.07	N	−0.74	N	1.23	MW	1.81	VW	−1.16	MD	
Aug.	−0.71	N	0.21	N	−0.46	N	−0.48	N	−0.78	N	−0.92	N	−0.69	N	0.38	N	1.89	VW	1.54	VW	
Sep.	−0.64	N	−0.84	N	−0.60	N	−0.26	N	−0.83	N	0.65	N	0.27	N	−0.31	N	0.08	N	2.48	EW	
Oct.	−0.68	N	−0.95	N	−0.57	N	−0.65	N	−0.40	N	−0.97	N	0.93	N	1.85	VW	0.24	N	1.20	MW	
Nov.	−0.57	N	−1.24	MD	0.02	N	−1.17	MD	0.51	N	−1.22	MD	0.77	N	1.57	VW	0.43	N	0.89	N	
Dec.	−0.83	N	0.01	N	−0.64	N	−1.07	MD	−1.20	MD	−0.27	N	1.28	MW	0.67	N	1.74	VW	0.33	N	
Notes.

N Normal

MW Moderate wet

VW Very wet

EW Extreme wet

MD Moderate drought

SD Severe drought

ED Extreme drought

Table 4 Z-score for discharges (ZSD) values.

Month	ZSD	ZSD	ZSD	ZSD	ZSD	ZSD	ZSD	ZSD	ZSD	ZSD	
	2008	Criteria	2009	Criteria	2010	Criteria	2011	Criteria	2012	Criteria	2013	Criteria	2014	Criteria	2015	Criteria	2016	Criteria	2017	Criteria	
Jan.	−1.27	MD	−0.44	N	−1.36	MD	0.41	N	−0.41	N	1.12	MW	−0.66	N	0.04	N	1.09	MW	1.46	MW	
Feb.	−0.46	N	−0.52	N	−0.51	N	−0.42	N	2.24	EW	1.48	MW	−0.57	N	−0.52	N	−0.34	N	−0.39	N	
Mar.	1.76	VW	−0.27	N	−0.43	N	1.79	VW	0.27	N	−0.59	N	−0.85	N	−0.80	N	−0.81	N	−0.07	N	
Apr.	1.68	VW	−0.29	N	−0.97	N	0.31	N	0.98	N	0.72	N	−1.04	MD	0.63	N	−1.16	MD	−0.86	N	
May	−0.85	N	0.40	N	0.72	N	−0.84	N	−0.57	N	2.26	EW	−0.48	N	−0.37	N	−0.85	N	0.60	N	
Jun.	1.44	MW	−0.39	N	2.25	EW	−0.49	N	−0.67	N	−0.19	N	−0.54	N	−0.59	N	−0.35	N	−0.48	N	
Jul.	0.16	N	−0.45	N	2.77	EW	−0.35	N	−0.16	N	−0.45	N	−0.70	N	−0.28	N	−0.12	N	−0.42	N	
Aug.	0.76	N	−0.64	N	2.22	EW	−0.20	N	−0.64	N	−1.02	MD	−0.40	N	0.94	N	−0.56	N	−0.49	N	
Sep.	0.36	N	−0.55	N	2.62	EW	−0.26	N	−0.59	N	−0.58	N	−0.49	N	−0.40	N	−0.59	N	0.47	N	
Oct.	−0.55	N	−0.65	N	−0.38	N	1.02	MW	−0.24	N	−0.85	N	1.31	MW	1.85	VW	−0.82	N	−0.71	N	
Nov.	0.46	N	0.57	N	0.97	N	−1.44	MD	−0.07	N	−1.87	SD	1.15	MW	0.65	N	−0.44	N	0.02	N	
Dec.	−1.20	MD	−0.49	N	1.22	MW	−0.34	N	0.76	N	1.24	MW	1.00	MW	−1.03	MD	−1.24	MD	0.07	N	
Notes.

N Normal

MW Moderate wet

VW Very wet

EW Extreme wet

MD Moderate drought

SD Severe drought

ED Extreme drought

The agreement between ZSP and ZSD indices reached 85.8%, with hydrological drought analyzed based on water discharge (ZSD), and meteorological drought analyzed based on rainfall (ZSP). Both indices tended to be negative during the dry season (April–September) indicating that the rainfall or discharge was below the average index. However, they were above the average index during the rainy season (October–March) (Table 5).

Table 5 Consistency between ZSP and ZSD.

Month/Year	2008	2009	2010	2011	2012	2013	2014	2015	2016	2017	%	
January	C	C	C	C	C	C	C	C	C	C	100	
February	C	C	C	C	IC	C	C	C	C	IC	80	
March	IC	C	C	IC	C	C	C	C	C	IC	70	
April	IC	C	C	C	C	C	C	C	C	IC	80	
May	C	C	C	C	C	IC	C	C	C	C	90	
June	C	C	IC	C	C	C	C	C	IC	C	80	
July	C	C	IC	C	C	C	C	C	C	C	90	
August	C	C	IC	C	C	C	C	C	C	IC	80	
September	C	C	IC	C	C	C	C	C	C	IC	80	
October	C	C	C	C	C	C	C	C	C	C	100	
November	C	C	C	C	C	C	C	IC	C	C	90	
December	C	C	C	C	C	C	C	C	IC	C	90	
Average	85.8	
Notes.

C Consistent (If the difference in drought class between ZSP and ZSD ≤ one class), for example:

ZSP Normal (N) ZSD Normal (N)/Medium Drought (MD)/Medium Wet (MW)

IC Inconsistent (If the difference in drought class between ZSP and ZSD >one class), for example:

ZSP Normal (N) and ZSD = Very Wet (VW)/Very Drought (VD)/Extreme Wet (EW)/Extreme Drought (ED)

Water availability and demand

Surplus and water deficits in the Krueng Jrue sub-watershed were calculated using the difference between water availability (discharges) and water demand (water irrigation + water for households). A positive difference between water availability and water demand indicates a surplus. On the contrary, a negative difference between water availability and water demand indicates a deficit (Table 6 and Fig. 4). Generally, there is a surplus of water availability during the rainy season (October–March) and a water deficit during the dry season (April–September).

Table 6 Deviation of water availability and demand in Krueng Jrue sub-watershed.

Year	2008	2009	2010	2011	2012	2013	2014	2015	2016	2017	
Jan.	−0.56	12.02	−1.99	24.89	12.41	35.54	8.51	19.14	35.07	40.62	
Feb.	−2.20	−2.80	−2.68	−1.95	21.52	14.76	−3.29	−2.91	−1.35	−1.82	
Mar.	10.85	0.17	−0.70	10.98	2.94	−1.61	−2.97	−2.74	−2.85	1.03	
Apr.	36.84	9.49	−0.02	17.70	27.06	23.41	−1.06	22.05	−2.70	1.47	
May.	−2.26	8.16	10.82	−2.25	−0.01	23.72	0.75	1.62	−2.44	9.71	
Jun.	1.91	−2.55	3.83	−2.84	−3.30	−2.15	−3.02	−3.17	−2.60	−2.93	
Jul.	−1.45	−2.78	4.18	−2.61	−2.21	−2.87	−3.43	−2.53	−2.22	−2.89	
Aug.	0.70	−2.01	3.45	−1.20	−2.06	−2.82	−1.64	0.91	−2.00	−1.88	
Sep.	9.52	−2.00	38.04	1.64	−2.51	−2.41	−1.28	−0.16	−2.56	10.77	
Oct.	0.06	−0.72	1.31	11.92	2.38	−2.30	14.14	18.21	−2.12	−1.32	
Nov.	32.50	33.88	39.33	6.73	25.27	0.79	41.61	34.86	20.08	26.38	
Dec.	6.36	16.28	39.97	18.33	33.57	40.28	36.83	8.63	5.62	23.91	
Notes.

Surplus (+sign) and deficit (-sign).

Figure 4 Surplus (+) and deficit (-) of water in the Krueng Jrue sub-watershed.

The monthly average water needs for irrigation and households were 5 m3 s−1 and 0.041 m3 s−1, respectively. There was a total water need of 5.041 m3 s−1 per month, and the average water availability (Table 2) from 2008 to 2017 in June (3.36 m3 s−1), July (3.16 m3 s−1), and August (4.19 m3 s−1) was unable to meet the water needs. A more detailed analysis of water availability for each year was conducted (Table 6 and Fig. 4). In 2009, there were five consecutive months during the gadu planting season (June, July, August, and September) with water availability lower than 3 m3 s−1, as well as from 2011 to 2012 (May to August) and from 2013 to 2017 (June to September). On the other hand, during the rendeng planting season (rainy season) in October 2009, 2013, 2016, and 2017, and February 2008-2011 and 2014–2017, the water availability was not enough to meet irrigation and household needs. Water deficit and surplus also occurred in both the rainy and dry seasons.

Discussion

Monthly discharges fluctuated depending on the amount of rainfall, the primary input of the Mock model. The rainfall-runoff model, known as the Mock model, was introduced by Mock (1973) based on a long-term study of rivers on the island of Java, Indonesia. Many Indonesian researchers have used the Mock model to analyze water availability in watersheds on islands outside Java. This model uses certain values for parameters that are specifically related to the proportion of surface soil that is not covered by vegetation (m), initial soil moisture (ISM), precipitation factor (PF), infiltration factor (IM), and flow reduction coefficient (Rc). In this study, the m, ISM, IF, and Rc values were 20%, 200 mm month−1, 0.4, and 0.6, respectively. A study of water availability based on satellite rainfall in the upstream Brantas watershed, Indonesia used the Mock model parameters, namely m = 30%, ISM = 250 mm month−1, IF = 0.75, and Rc = 0.85 (Maulana, Suhartanto & Harisuseno, 2019). Research on hydrologic modeling for tropical watershed monitoring and evaluation used the values m = 30%, ISM = 100, IF = 0.5, and Rc = 0.85 (Setyawan, Lee & Prawitasari, 2016). This shows that, due to differences in climate, land use, and soil types, the parameter values of the Mock model vary between watersheds.

The discharges generated from the Mock model were influenced by rainfall, evapotranspiration, land cover, and soil types, and usually, the higher the rainfall, the higher the discharge produced. Taking this into account, a high evapotranspiration value will cause a decrease in discharge, although it is not very significant. A reduced land cover increases surface runoff, otherwise, the infiltration process will be hampered (Basri, Manfarizah & Chandra, 2021). Land use and soil types influence the availability of water in a watershed (Basri et al., 2022; Alayani, Sugianto & Basri, 2021; Ihsan, Rusdi & Basri, 2021), where soil types with high clay content have a higher water holding capacity than sandy soils.

Rainfall has a strong influence on discharges in a watershed. Therefore, meteorological drought represented by a ZSP value can be analyzed using the hydrological drought represented by a ZSD value. In this study, the consistency between the ZSP and ZSD indices reached 85.8%, indicating that the meteorological droughts analyzed based on rainfall (ZSP) and hydrological droughts analyzed based on water discharge (ZSD) were in line. However, there was 14.2% inconsistency that may have been influenced by watershed biophysical conditions that affect water balance components such as evapotranspiration, land cover, infiltration, surface runoff, and subsurface flow. Regarding the rainfall-runoff model, the model’s reasonability, especially the tank model’s, can be increased by considering soil types, land use types, rainfall, and actual discharges (Basri, 2013).

Analysis between water availability and demand for irrigation and household needs is very important when anticipating water shortages. The need for irrigation water in Indonesia is influenced by the growing season. The Krueng Jrue sub-watershed has two rice growing seasons: the rendeng planting season (October–February) during the rainy season and the gadu planting season (May–September) during the drought season. There is a two-month bera period (March–April) when farmers rest and provide opportunities for the land to recover. Usually, during the rendeng planting season (rainy season), water availability can meet irrigation and household needs, but is often insufficient during the gadu planting season (dry season). However, in the last 10 years, the water availability could not meet the irrigation and household needs for specific months, such as in June, July, and August. In 2009, there were five consecutive months in the gadu planting season (June, July, August, and September) as well as from 2011 to 2012 (May to August) and from 2013 to 2017 (June to September) with water availability lower than 3 m3 s−1. In the rendeng planting season (rainy season) in October 2009, 2013, 2016, and 2017, as well as in February (2008 –2011 and 2014 –2017), the water availability was not enough for the irrigation and household needs. Although water surplus was found in the rainy and dry seasons, water deficit also occurred in both the rainy and dry seasons. These findings indicate the uncertainty of water availability during the rainy season and dry season which can affect the amount of water available for irrigation and household water needs. Therefore, a shift in the growing season for paddy and secondary crops should be considered by the local government and farmers in the future.

Hydrological drought leading to inadequate water for irrigation and households, especially in the dry season, can occur annually. Various technical and nontechnical hydrological drought mitigation efforts can be implemented as part of a sustainable watershed management system to mitigate hydrological drought (Asdak & Supian, 2018). The technical method that can be implemented in the Krueng Jrue sub-watershed includes maintaining the function of the irrigation network (Wibowo, Wardoyo & Edijatno, 2018); building water traps, terraces, and water retention ponds (Pramono & Savitri, 2017); and maintaining conservation areas, especially the upstream as a natural reservoir. This will increase infiltration into the soil and reforestation in the upper catchment area by planting trees to increase spring water discharge and water availability (Cao et al., 2010). The nontechnical methods can be carried out by enforcing some available regulations to prevent forest to non-forest land conversion, conducting soil and water conservation, river development, and river water damage control, as well as monitoring and evaluating watershed management.

Conclusion

This study concludes that the Z-score statistics method can be used to analyze meteorological drought based on rainfall (ZSP) and hydrological drought based on discharges (ZSD) with the consistency between the ZSP and ZSD indices reaching 85.8%. The correlation between observed discharges and calculated discharges generated using the Mock model showed a very significant regression coefficient (r = 0.95**) and coefficient of determination (R 2 = 0.90**). We recommend Mock model parameters for the proportion of soil surface that is not covered by vegetation (m), infiltration factor (IF), initial soil moisture (ISM), and flow reduction coefficient (Rc) of 20%, 0.4, 200 mm month−1, and 0.6, respectively. However, the use of this Mock model for other watersheds requires calibrating these parameters because there are differences in the climatic and biophysical conditions of the watersheds.

In general, this area had a normal (N) Z-score for precipitation (ZSP) class with 90 events (75%), with six to 11 moderate wet (MW), very wet (VW), extreme wet (EW), and moderate drought (MD) events (5−9.2%), and 0 events (0%) for severe drought (SD) and extreme drought (ED). Furthermore, this region had a normal (N) Z-score for a discharge (ZSD) class with 89 events (74.2%); four to 11 moderate wet (MW), very wet (VW), extreme wet (EW), moderate drought (MD) events (3.3−9.2%); and one severe drought (SD) event (0.8%).

The ZSP and ZSD indices in the dry season (April–September) tended to be negative, indicating that rainfall or discharge was under the average index. On the contrary, they tended to be positive and above the average index for the rainy season (October–March). There was a surplus of water for irrigation and household water needs during the rainy season and a deficit for gadu planting in the dry season. On the other hand, the availability of water during the rendeng planting season (rainy season) in October 2009, 2013, 2016, and 2017, as well as in February (2008–2011 and 2014–2017), was insufficient for household and irrigation. This observation is probably due to global climate change factors that need to be substantiated by further research. To overcome the water shortage, it is necessary to rearrange planting schedules, save water, improve irrigation networks, and guide the community and government regarding the importance of maintaining the condition of the catchment area.

Supplemental Information

Supplemental Information 1 Drought and water availability analysis in the Krueng Jreu Sub-Watershed

Click here for additional data file.

Supplemental Information 2 Surplus and deficit of water in Krueng Jreu Sub-Watershed

Click here for additional data file.

The authors gratefully acknowledge the Krueng Aceh Watershed Management Board; the Blang Bintang Meteorological, Climatological, and Geophysical Agency; and the Center of River Basin Sumatera-I, for providing essential data regarding land use, soil type, climatology, and actual discharges.

Additional Information and Declarations

Competing Interests

Author Contributions

Data Availability

The authors declare there are no competing interests.

Hairul Basri conceived and designed the experiments, performed the experiments, analyzed the data, prepared figures and/or tables, authored or reviewed drafts of the article, and approved the final draft.

Sufardi Sufardi conceived and designed the experiments, authored or reviewed drafts of the article, and approved the final draft.

Helmi Helmi performed the experiments, prepared figures and/or tables, and approved the final draft.

Syakur Syakur performed the experiments, authored or reviewed drafts of the article, and approved the final draft.

Sugianto Sugianto performed the experiments, authored or reviewed drafts of the article, and approved the final draft.

Azmeri Azmeri analyzed the data, authored or reviewed drafts of the article, and approved the final draft.

Helmi Helmi conceived and designed the experiments, analyzed the data, prepared figures and/or tables, and approved the final draft.

The following information was supplied regarding data availability:

The raw data is available in the Supplemental Files.

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
