# Peer review of "Drought and water availability analysis for irrigation and household water needs in the Krueng Jrue sub-watershed"

_PeerJ, doi:10.7717/peerj.14830_

## Round 0.1 · original submission · Major Revisions

Dear Authors,

As you shall see that reviewers have now commented on your manuscript. They are suggesting a major review. It can be noted that all of the reviewers have pointed out issues in communicating your ideas related to rationale, problem statement, aims and objectives, and the results and discussion. Therefore it is suggested to considerably revise your manuscript in structure, coherence, and clarity. Further, the discussion section must include the limitations of this study and the future scope.

I hope you will do your best to revise this manuscript. While revising, do not forget to check out the annotated files attached to the reviewer reports.

I am looking forward to receiving your review in time.

Best regards
Gowhar Meraj

·

Basic reporting

No further comments. All detailed comments have been attached to the reviewed manuscript

Experimental design

Over all is acceptable with improvement of data/model validity as mentioned in the reviewed manuscript

Validity of the findings

Over all is valid including the important findings

Additional comments

See the detailed comments over the reviewed manuscript

Reviewer 2 ·

Basic reporting

The English needs significant improvement. The authors may need an English correction software or service.

More literature is needed to support some of the claims made in the manuscript and to enrich the introduction and discussion sections.

There was no mention of hypothesis. I presume this is applied research.

Experimental design

The authors need to improve on language relating to the novelty of the research. The research question is not well defined, and the authors did not indicate which knowledge gap the research will address. The only gap statement is that the two methods have not been applied in the study area. I recommend that the authors redefine their study problem to indicate whether it is a knowledge or data gap.

Validity of the findings

The authors need to link their findings to the gap statement.

Additional comments

Line 19: I suggest that the authors revise this sentence: the opening clause is probably not necessary, it implies. Additionally, many countries and globally seem redundant. Also remove the phrase including Indonesia and make this a separate sentence that contextualizes the global issue.
Lines 36-37: confusing sentence. Please revise.
Line 39: I suggest that the authors replace vice versa with an explanation of what they observed in the rainy season.
Line 44: Ditto
Lines 47-48: Please revise. Example: This observation is probably due to the influence of global climate variables on the community. However, the climate change claim needs to be substantiated.

Introduction section: the major issue with the introduction is that the writing is poor. I recommend that the authors seek some writing service to help them improve the quality of their manuscript. See examples below:

Line 53: See my suggestion: The impacts of climate change include drought, flooding, and water scarcity.
Line 53-54: I don’t think it is necessary to point to a country or an entity to address the impacts of climate change. I suggest that the authors stick to third person style of writing. For example: the climate problem requires and understanding of trends the in hydrological dynamics based on the current biophysical conditions of a watershed to allow better preparedness and response.
Line 56: meteorological and hydrological drought and water availability – are these terms used correctly?
Lines 104-106: My understanding – the authors mean that the Z-score statistical method and the Mock model have been used elsewhere but not in Krueng Jrue Sub-Watershed. Assuming that is the case, and apart from the study area, what else is the novelty of this research?
Line 187: Why did the authors select these methods as compared to other existing methods?
Line 189: Please revise your titles to reflect the contents of the paragraphs.
Line 190: The opening paragraph is misleading. The title says discharge using the mock method, but the paragraph speaks about rainfall and seasons.
Line 199: The authors keep throwing in the term climate change, but they have not provided sufficient scholarly materials to back up the ideas that mention it.
Line 200: What does …originated from the Mock model mean?
Line 209: What does monthly debit mean?
Line 211: I suggest that the authors avoid using vice versa
Line 243: The authors use vice versa a lot. Is this a formal style of writing?
Line 255: Please explain with scholarly backing how your findings are related to global climate change.

Discussion section: the discussion section is mainly a summary of the authors' work. The discussion needs to be heavily cited. This is where the authors reiterate their study findings and then compare to the findings of precious work under similar settings. Please take each finding and discuss (i) the key highlights of the finding (ii) what does the finding mean or imply? Why is the finding the case? (iii) which previous works corroborate the finding (iv) which previous works found results that contradict your finding, and why? (v) how scalable is the result, how does it contribute to the body of knowledge?

The conclusion needs minor revisions based on recommendations made above.

Annotated reviews are not available for download in order to protect the identity of reviewers who chose to remain anonymous.

Reviewer 3 ·

Basic reporting

The manuscript is written in clear and good english. some minor typo can be seen, e.g., line 108



Literature review is enough

Figures are clear but not in good qulity

Tables are good

results are relevant and well described

Experimental design

Line 47-48: the worldwide climate change is a vague term. Climate change can have different impacts in different locations and places around the world.

Validity of the findings

The results are useful for assessing the droughts in the region

---

## Round 0.2 · Minor Revisions

The authors strongly suggested improving the language of the manuscript. It is very important that the manuscript is proofread by an English-speaking expert. It has a lot of language issues. Therefore please revise your manuscript accordingly and submit the revised manuscript as soon as possible.

Best regards,
Gowhar Meraj

·

Basic reporting

Basic reporting has been met sufficiently in line with the journal standard

Experimental design

Experimental design has also met with most hydrological research methodology

Validity of the findings

The findings are valid, and in according to standard hydrological research

Additional comments

No further additional comments considering that the revised version has sufficiently responded my first comments for improvement

Reviewer 2 ·

Basic reporting

The authors have agreed to my suggestions. Therefore, I believe that the article is publishable. However, there are minor grammatical errors that editor may want to check.

Experimental design

Looks good

Validity of the findings

Looks good

---

## Round 0.3 · accepted · Accept

Thank you for incorporating all the suggestions made by the reviewers and the editor.